# 4000ers of the Alps–So beautiful, so dangerous: An analysis of falls in the Swiss Alps between 2009–2020

**Benedikt Gasser** *, **Fabian Schwendinger**

Department for Sport, Exercise and Health, University of Basel, Basel, Switzerland

☯ These authors contributed equally to this work.
* Benediktandreas.gasser@unibas.ch

**Data Availability Statement:** All relevant data are within the paper and its Supporting Information files. A minimal anonymized data set is provided to allow for replication of the figures and calculations.

## Abstract

### Background

This study aimed to analyze falls regarding their demographic characteristics, severity, frequency over time, and the localization of injuries while high-altitude mountaineering in the Swiss Alps.

### Methods

Data on fall-related emergencies during mountaineering between 2009 to 2020 from the Swiss Alpine Club central registry were analyzed retrospectively. The variables age, sex, time of occurrence, severity of an event quantified by NACA-Score (National Advisory Committee for Aeronautics Score), and injury localization were examined descriptively. Changes in injury severity, number of total emergencies, and fatal emergencies over time were analyzed using linear regression models.

### Results

Out of 1347 (28.7%) victims of fall-related emergencies, 1027 were men (76.2%) and 320 (23.8%) women. Around 70% of the cases happened during summer in July and August. The mean age was 49.9 ± 14.9 years for men and 51.3 ± 14.4 years for women with no between-sex difference. Higher NACA-Scores were found in men than women (3.6 ± 2.2 vs. 3.1 ± 1.6; $p < 0.01$). Approximately 80% of all victims originated from the countries close to the Alps (Switzerland, Germany, Italy, France, and Austria). There was a slight decrease of total cases ($R^2 = 0.104$) and fatal cases over time ($R^2 = 0.183$). NACA-Scores decreased over time ($R^2 = 0.168$). Likewise, risk aversion decreased over time as the severity of emergencies decreased. Injuries occurred predominantly at the extremities (90%). Out of all cases, 228 fatal emergencies (16.9%) could be identified of which 82 occurred while climbing one of the classic 4000ers.

### Conclusions

The decrease of the number and severity implies that security standards of the average alpinist have in tendency increased. Nevertheless, the high number of emergencies on

**Funding:** The authors received no specific funding for this work.

**Competing interests:** The authors have declared that no competing interests exist.

classic 4000ers implies that despite the potentially improved security standards, many tours on famous mountains still have high requirements in terms of alpine skills.

## Introduction

Mountain sports activities enjoy increasing popularity and rising numbers of executants [1–3]. In 2001, the number of tourists visiting altitudes above 2000 m was estimated at 40 million people per year in the Alps and 100 million high-altitude tourists worldwide [1]. For the Swiss Alps, it is assumed that an estimated number of 150,000 persons engage in high-altitude mountaineering per year [4]. Despite the beauty of the Swiss Alps, risks while mountaineering exist despite all technological progress.

The fall is a famous and central cause of death while high-altitude mountaineering [5] and tragedies already started with the first ascent of popular peaks such as the Matterhorn [2, 6]. Seven persons were on the peak, however during the descent four died due to a rope rip [2, 6]. Further tragedies are well known from the large north walls such as the Eiger-Nordwand or the Grandes Jorasses [2, 6]. Today, it is hardly comprehensible with what boldness big walls were successfully completed in the past [2]. At the stand, well-secured with drilled bolts or ice screws, equipped with excellent material and familiar with state-of-the-art safety technology, successfully completing difficult routes is nowadays definitely possible at a lower risk [2]. Nevertheless, falls still have the highest relevance and even top alpinists are not filed against them as for instance the tragedy of Ueli Steck shows [7, 8]. Apparently, the weather was good at the time of the fall and the route was also not known to be technically extremely demanding [7]. It seemed to be a routine exercise for an exceptional alpinist like Ueli Steck [7]. Nevertheless, even the best are not immune to falling and the danger of falls in high-altitude mountaineering is constantly present [2, 7, 8]. In consequence, the prevention of these was repeatedly empathized by the SAC (Swiss Alpine Club) [2].

Difficult parts are now often secured with drill hooks on the classic routes and the technical developments around the use of securing material is continuing enormously while parallel increasing security [2]. Focusing on the effective causes of falls diverse reasons can be mentioned: the breaking out of a rock, a slippery stand or the good grip that simply does not come are just a few causes for falls in the high alps [2]. However, from a systematic point of view, there is, unfortunately, little data on the characteristics of falls while high-altitude mountaineering in contrast to for example mountain hiking [3, 9]. Therefore, the aim of this study was to analyze the characteristics of falls during high-altitude mountaineering in the Swiss Alps between 2009 and 2020.

## Materials and methods

### Study population

This study included all mountain emergency cases involving high-altitude mountaineering that were documented in the SAC central registry between 2009 and 2020. The registry contains data from the Swiss Air Rescue Service (REGA), Air Glaciers Lauterbrunnen, Air Glaciers Sanenland, Register SAC, KWRO (Kantonale Walliser Rettungsorganisation), Snow and Avalanche Research Institute Davos, and the cantonal police registers. This study was conducted according to the local regulatory requirements respectively the declaration of Helsinki (1964) and its further amendments. Since original data analyzed was anonymous, no consent for participation was necessary and a waiver was obtained by the ethics committee of North-western

and Central Switzerland. All data were fully anonymized before we accessed them. The term 'mountain emergency' covers all events involving alpinists requiring the help of mountain rescue services and alpinists being affected by subjective and objective mountain hazards [10–12]. This also applies to illnesses and evacuations of uninjured mountaineers. Each mountain emergency included the emergency number used, date, rescue organization, event, place, canton, activity, NACA-Score (National Advisory Committee for Aeronautics Score), nationality, date of birth, sex, place of residence, coordinates, and a short report (Table 1) [13, 14].

## Data preparation

In the first step, all mountain emergencies were classified according to their cause. Cases that occurred due to falls were subsequently analyzed in detail in terms of age, sex, time of occurrence, severity of the event quantified by NACA-Score (National Advisory Committee for Aeronautics Score), and injury localization. This was followed by a detailed data analysis for the missing entries. For further analyses of age, a substitution method (mean value imputation as missing values were less than five percent) was used [15].

## Statistical analyses

Descriptive statistics were calculated for age and NACA-Score for women and men, respectively. To analyze potential differences in age between sexes, a two-sided heteroscedastic t-test was performed. Normality was assessed graphically using quantile-quantile plots. Since NACA-Scores were not normally distributed, a Mann-Whitney U test was used to analyze between-sex differences. Changes over time were examined using linear regression with the calculation of the coefficient of determination ($R^2$). Furthermore, the model of risk-bearing proposed by Arrow (16) & Pratt (17) was applied. Starting with a polynomic regression of second degree of NACA-Score over time, the absolute aversion of risk (AAR) = $-$ f'(t)/f''(t) could further be calculated to approximate risk aversion. Analyses were done using Microsoft Excel (Microsoft Inc., Redmond, WA, USA) and SPSS statistics (Armonk, New York, USA).

## Results

Out of 4687 cases of emergencies in the observational period from 2009–2020, a total of 1347 fall-related cases were identified. The distribution of falls over the year showed that emergencies mainly occurred in the summer months (Fig 1). Of this sample, 1,027 (76.2%) were men and 320 (23.8%) were women. The mean age was 50 ± 15 years for men and 51 ± 14 years for

**Table 1. NACA-Score (National Advisory Committee for Aeronautics Score) [13, 14].**

| | |
|---|---|
| NACA 0 | No injury or disease. |
| NACA I | Minor disturbance. No medical intervention is required (e.g., slight abrasion). |
| NACA II | Slight to moderate disturbance. Outpatient medical investigation but usually no emergency medical measures necessary (e.g., fracture of a finger bone, moderate cuts, dehydration). |
| NACA III | Moderate to severe but not life-threatening disorder. Stationary treatment required, often emergency medical measures on the site (e.g., femur fracture, milder stroke, smoke inhalation). |
| NACA IV | Serious incident where rapid development into a life-threatening condition cannot be excluded. In the majority of cases, emergency medical care is required (e.g., vertebral injury with neurological deficit, severe asthma attack, drug poisoning). |
| NACA V | Acute danger (e.g., third grade skull or brain trauma or severe heart attack). |
| NACA VI | Respiratory and or cardiac arrest. |
| NACA VII | Death. |

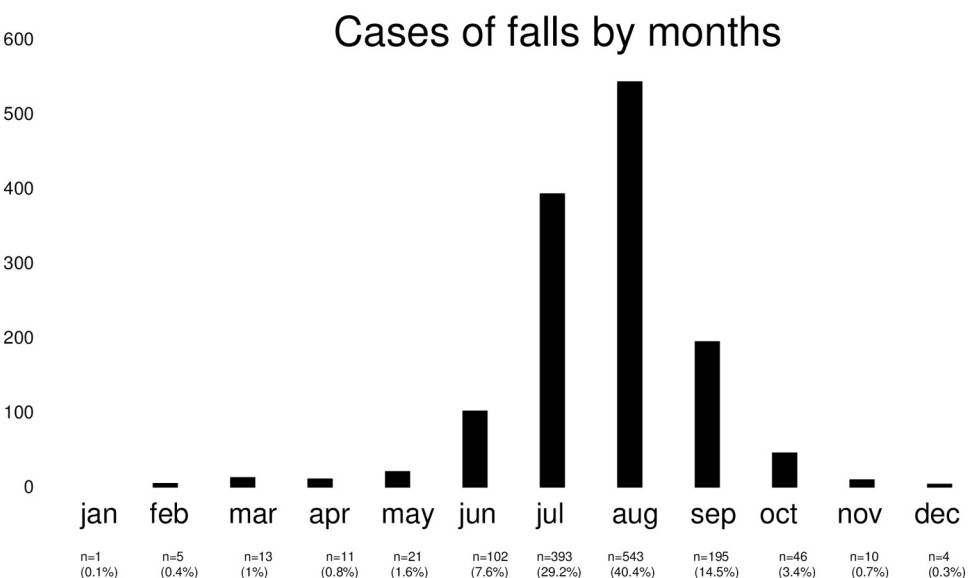

**Fig 1. Distribution of falls during high-altitude mountaineering throughout the year–around 70% occurred in the two summer months July and August.**

women, with no statistical difference between sexes (p = 0.14). Victims of fall-related emergencies originated from Switzerland (49.2%), Germany (18.3%), Italy (5.9%), France (5.3%), Austria (3.7%), the UK (2.7%), the Netherlands (2%), Poland (2.3%), Czech Republic (1.7%), Spain (1%), Belgium (1%), Japan (1%), the U.S. (0.5%), and other countries (3%). Taken together, the countries from the Alps, Switzerland, Germany, Italy, France, and Austria, covered around 85% of all cases. Regarding the severity of the injuries suffered, mean NACA-Scores were significantly different between men and women (3.6 ± 2.2 vs. 3.1 ± 1.6; p < 0.01).

A total of 228 fatal fall-related emergencies were detected (16.9% of all fall-related cases) with 195 (85.5%) alpinists being men and 33 (14.5%) women. The mean age was 51 ± 14 for men and 49 ± 14 for women, whereby no statistical difference between sexes was detected (p = 0.37). Victims of fatal emergencies originated from Germany (29.8%), Switzerland (27.6%), Italy (11.8%), France (6.1%), Austria (3.5%), Czech Republic (3.5%), Japan (2.6%), and other countries (2%). Eighty-two fatal events (36.0%) took place at one of the classic 4000 m mountains [18] with 17 (7.5%) at the Matterhorn and 12 (5.2%) at the Bernina.

Data on the injury localization were available for 482 (35.8%) cases. There was a relatively low share of head and trunk injuries of only around 5% each, whereas the upper extremities with close to 50% and lower extremities with around 40% were much more likely to be injured (Table 2).

Based on the case reports it is to mention that in 971 cases alpinists were not secured by a rope or where falling alone and hold by the companions. Furthermore, pairs of two alpinists were identified in 266 cases, in 67 cases groups of three alpinists, in 22 cases groups of four alpinists and in 21 cases groups of five alpinists or more.

The development of both the number of total cases ($R^2 = 0.104$) and fatal cases ($R^2 = 0.183$) was characterized by a slight decrease over time (Fig 2A and 2B). Furthermore, the severity of emergencies decreased over time ($R^2 = 0.168$) (Fig 2C). Using the concept of Arrow and Pratt [16, 17], we tried to quantify the development of risk aversion over time, whereby stating a direct relationship between severity of an injury and risk aversion. Therefore a polynomial regression of second degree of development of NACA-Score over time was used, yielding

**Table 2. Localization of injuries from 482 (35.8%) emergency cases attributed to anatomical location (multiple choices were possible).**

| Anatomic region | Body part | n | % |
|---|---|---|---|
| **Lower extremities** | Foot | 115 | 16.6 |
| **n = 289** | Ankle | 13 | 1.9 |
| **41.80%** | Ankle joint | 19 | 2.7 |
| | Shank | 30 | 4.3 |
| | Knee | 51 | 7.4 |
| | Thigh | 3 | 0.4 |
| | Leg | 58 | 8.4 |
| **Upper extremities** | Finger | 11 | 1.6 |
| **n = 330** | Hand | 25 | 3.6 |
| **47.80%** | Arm | 228 | 33 |
| | Shoulder | 66 | 9.6 |
| **Trunk** | Rip | 15 | 2.2 |
| **n = 36** | Chest | 4 | 0.6 |
| **5,2%** | Stomach | 5 | 0.7 |
| | Back | 12 | 1.7 |
| | Spine | 2 | 0.3 |
| **Head** | Head and face | 34 | 4.9 |
| **4.90%** | | | |
| | **Total** | **691** | **100** |

$NACA_t = -0.01 * time^2 + 0.13 * time + 3.20$ ($R^2 = 0.384$). In addition, the AAR was 4.1 for the year 2009 and 6.9 for the year 2020. As negative, respectively smaller values imply risk pleasure and positive values risk aversion, it is detectable that risk aversion decreases over time with the average rate per year calculated as: ([AAR $(t_{12})$ / AAR $(t_1)$]1/12–1) * 100 = 4.3%. To consider, this statement is biased by many factors such as e.g. better training or equipment.

## Discussion

The aim of this study was to analyze falls during high-altitude mountaineering in the Swiss Alps. The central findings of this twelve-year retrospective analysis were that the majority of accidents happen in the summer months July and August with severe injuries and fatal accidents occurring more frequently in men than women. With 90% of all injuries, the extremities are the most frequently injured body region. Furthermore, there was a slight decrease in both the number of fall-related emergencies and the severity of injuries apparent over the past 12

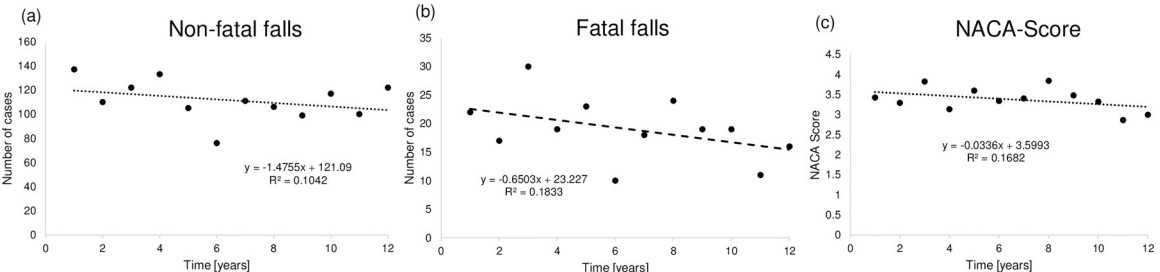

**Fig 2.** (a) Development of all falls over the observational period (b) Development of fatal falls over the observational period (c) Development of NACA-Score over the observational period. The score indicates the severity of injuries. Abbreviations: NACA-Score, National Advisory Committee for Aeronautics Score.

years. These findings may advance alpine sports epidemiology and form an important basis for fall-related injury prevention in high-altitude mountaineering.

As most tours to the top of 4000ers are absolved best in the summer months [18], the finding indicating a high share of emergencies occurring in these months seems valid and correlates with previous research [19]. It seems hence suggestive that mountain safety campaigns (e.g., sicher-bergwandern.ch) should aim to achieve the highest visibility in these months.

Male sex may be associated with a higher prevalence of both non-fatal and fatal emergencies during mountaineering. This is contrary to the sex distribution seen in non-fatal falls in mountain hikers in the Austrian alps [9]. The well-described elevated risk-taking behavior in males might be one possible explanation [20]. Yet, a possible underrepresentation of female mountaineers could also have impacted the distinct between-sex difference.

The mean age of victims of non-fatal and fatal falls in the current study was around 50 years for both sexes. This is six years lower compared to victims of non-fatal falls during mountain hiking [3] but still 8 years older than the general population of mountain hikers [21]. As cardiorespiratory fitness decreases with advancing age [22], older alpinists may be more easily over-challenged by environmental factors (i.e., pronounced hypoxia, solar radiation, etc.) and difficult trail conditions, posing a greater physiological strain compared to mountain hiking [23]. Advancing age may thus be associated with an increased number of emergencies during mountaineering.

When further focusing on the nationality of victims, the portion of victims with Swiss nationality was markedly smaller in fatal falls compared to non-fatal falls (27.6% vs. 50.0%). This supports the argument that alpinists being unaware of the local terrain are at greater risk for fatal events. It may be assumed that Swiss alpinists have developed a special security sensorium due to due living close to the mountains for their whole life, allowing them to assess mountaineering risks more adequately [10–12].

The location of injuries seems to be important information for the advancement of prevention strategies as well as mountaineering equipment. It was evident from our data that the injuries were relatively evenly distributed between the lower (around 40%) and upper extremities (nearly 50%). The relatively low frequency of head injuries with only 5% might be because nowadays alpinists often wear a helmet. The slightly higher prevalence of injuries at the upper extremities may be explained by the center of gravity predisposing to fall onto the upper extremities or the attempts of alpinists to catch the fall. Consequently, injury prevention should focus on the extremities.

From an epidemiological perspective, analyzing changes in the number of emergencies over time is important to examine whether prevention strategies and advances in mountaineering equipment are successful. Our data indicated a slight decrease of cases over the observational period for both fatal and non-fatal emergencies (Fig 2A and 2B). Furthermore, a decrease in the severity of emergencies was detected over time (Fig 2C). Reasons for these findings could be the higher standard of equipment [2]. This may firstly be due to enhanced personal equipment and secondly better equipment on the routes. Furthermore, the advances in GPS technology which is nowadays used by many alpinists may allow emergency services to act in a faster and more efficient manner. Long search actions with potentially further harming have probably become less likely yielding to a more secure sport in general. Finally, the decreasing number of injuries over time may also reflect the success of ongoing injury prevention campaigns.

Interestingly, the mean NACA-Score of men was slightly higher compared to women implying more severe injuries in male alpinists. This is supported by findings based on a similar dataset of the Alpine Club of Canada [19]. As mentioned before, the greater risk-taking behavior in men may be associated with the increased injury severity [20]. The fact that a

substantially greater percentage of fatal accidents was detected in men versus women, indicating a greater risk aversion in the latter backs this hypothesis.

A significant number of fatal cases occurred at the Matterhorn and the Piz Bernina. When interpreting this finding, it should however be taken into consideration that these mountains are very popular and thus more frequently visited. The likelihood for a fatal accident is in consequence larger. Yet, this argument probably only matters to some extent as a special type of alpinists may attempt to climb these popular mountains with a less developed risk sensorium.

Focusing on the likelihood of an event, it was estimated that 150,000 alpinists are active in the high mountains every year. This implies that around one in a thousand alpinists has an emergency due to a fall [4]. This is a relatively low prevalence rate, but it must be considered that the statistics only include more severe cases that are recovered by the organized emergency organizations [3]. Falls that do not cause the rescue teams to move out are consequently not recorded. The same applies to alpinists who self-refer themselves after a fall to a general practitioner or an emergency department. Another limitation was that reports about the emergencies are commonly short for organizational reasons, not always providing information about e.g., the location of the injury.

## Conclusions

To conclude, we found that most falls in mountaineering occur during the summer months. Male sex may be associated with both higher prevalence and severity of fall-related injuries. Advancing age may further have an impact on the prevalence of injuries. The extremities are injured most often and should therefore be moved into the spotlight. Recent efforts for improving mountaineering equipment and creating awareness for hazards in the mountains seem to work out. The findings of the present study have important implications for injury prevention and form a necessary basis for future research examining risk factors for fall-related injuries in mountaineering. Finally, from a practical point of view, we especially encourage male and/or older alpinists to perform adequate planning of the tour under consideration of the individual fitness status.

## Supporting information

**S1 Data.**
(XLSX)

## Acknowledgments

Special thanks goes to the members of the Geb Spez Abt. 1 for their constructive feedback.

## Author Contributions

**Conceptualization:** Benedikt Gasser.

**Data curation:** Benedikt Gasser.

**Formal analysis:** Benedikt Gasser.

**Funding acquisition:** Benedikt Gasser.

**Investigation:** Benedikt Gasser, Fabian Schwendinger.

**Methodology:** Benedikt Gasser.

**Project administration:** Benedikt Gasser.

**Resources:** Benedikt Gasser.

**Software:** Benedikt Gasser.

**Supervision:** Benedikt Gasser.

**Validation:** Benedikt Gasser.

**Visualization:** Benedikt Gasser.

**Writing – original draft:** Benedikt Gasser.

**Writing – review & editing:** Fabian Schwendinger.

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
