## [Decision Letter · Decision Letter 0]

24 Nov 2021

PONE-D-21-245014000ers in the Alps – so beautiful, so dangerous - an analysis of falls in the Swiss Alps between 2009-2020PLOS ONE

Dear Dr. Gasser,

Thank you for submitting your manuscript to PLOS ONE. After careful consideration, we feel that it has merit but does not fully meet PLOS ONE’s publication criteria as it currently stands. Therefore, we invite you to submit a revised version of the manuscript that addresses the points raised during the review process.

We look forward to receiving your revised manuscript.

Kind regards,

Zsolt J. Balogh, MD, PhD, FRACS

Academic Editor

PLOS ONE

Journal Requirements:

Reviewers' comments:

Reviewer's Responses to Questions

**Comments to the Author**

1. Is the manuscript technically sound, and do the data support the conclusions?

Reviewer #1: Yes

Reviewer #2: Yes

2. Has the statistical analysis been performed appropriately and rigorously? 

Reviewer #1: Yes

Reviewer #2: Yes

3. Have the authors made all data underlying the findings in their manuscript fully available?

Reviewer #1: Yes

Reviewer #2: Yes

4. Is the manuscript presented in an intelligible fashion and written in standard English?

Reviewer #1: Yes

Reviewer #2: No

5. Review Comments to the Author

Reviewer #1: Thank you for asking me to review this manuscript about an analysis of falls on the 4000 peaks in the Alps. The manuscript is original and has sound statistical analysis. It is easy to read and follow.

However a couple of points must be raised to make this manuscript more meaningful.

1. What other data is collected in the Swiss alpine club registry?

2. Do the authors have data on which mountain, route taken (classic vs modified, difficulty), weather conditions, skill set of mountaineering party (tourist vs experienced), number of people in the mountaineering group vs solo, equipment and safety gear etc.

3. Upgrades done on specific routes, some of the classic routes have minimal drill bolts and only at difficult sections to not take the experience for the mountaineer

4. The anatomic location should be more specific. What do the authors mean by ankle vs ankle joint? Shank? Leg? Maybe something similar to below is more adequate?

a. Foot fractures/injuries

b. Ankle fractures/injuries

c. Tibia and fibula fractures

d. Knee injuries

e. Femur fractures

f. Hip fractures/injuries

g. Pelvic fractures

h. Soft tissue injuries in any lower limb location

5. Similar for upper extremity and trunk. The anatomic location/injury needs to make sense! Arm?

6. How many patients needed hospitalization?

7. How many patients needed an operation?

8. Where they any falls that could have been prevented? No equipment, wrong equipment etc.

Reviewer #2: The article is a simple descriptive study which might allow useful formulation of public policy or approaches to rescue and therapy.

It is worth publishing.

Some of the translation into English should be improved for clarity.

line 58 "absolved " would be better written as "successfully completed"

line 60 "absolving " would be better written as "successfully completing" another option would be the climbing jargon " to summit"

line 223 patients who "self confine" the best English equivalent would be patients who "self present" to general practitioners etc.

In table 2 injuries to the trunk "rip" is not a common medical term. I don't know what that means. It may be a translation for laceration?

In the results section "risk aversion" is discussed. This is a psychological concept and may not explain the decrease in accident and injury rates. The change might be due to better training , better equipment or varying numbers of climbers going up mountains.

6. PLOS authors have the option to publish the peer review history of their article (what does this mean?). If published, this will include your full peer review and any attached files.

Reviewer #1: No

Reviewer #2: **Yes: **Dr Conrad Loten

---

## [Author Response · Author response to Decision Letter 0]

18 Jan 2022

Point-by-point response to reviewers

We thank the reviewers, for taking the time and providing their thoughtful and important feedback to our manuscript. Please find below the point-by-point responses to the reviewers’ comments. Text in italic are the reviewers’ comments, normal text is our response to the comments, and text in red is what was edited in the manuscript. 

To make sure the manuscript is written in a clear, correct, and unambiguous manner, a thorough spelling check was performed. We formatted the manuscript according to the PLOSE ONE style requirements and included an additional statement concerning the anonymity of the analyzed data (L. 89-90). Finally, we specified in the data availability statement, where the minimal data set underlying the results described in the manuscript can be found.

Reviewer 1

Reviewer 1 Comment 1: What other data is collected in the Swiss alpine club registry?

Authors’ response 1: The registry further includes information about the emergency number used to make the call, date, rescue organization, event, place, canton, activity, place of residence, and coordinates. This was mentioned in L. 138-142.

Reviewer 1 Comment 2: Do the authors have data on which mountain, route taken (classic vs modified, difficulty), weather conditions, skill set of mountaineering party (tourist vs experienced), number of people in the mountaineering group vs solo, equipment and safety gear etc.

Authors’ response 2: Thank you for this remark. In some but not all cases, information about weather conditions are available. However, this parameter is not routinely documented and is thus difficult to analyze. Detailed information about the route taken are usually not available. Finally, it is known whether the victims were mountaineering in a group or alone. However, we included an additional comment of the group sizes during mountaineering in the results section in the manuscript (see L. 147-150).

Reviewer 1 Comment 3: Upgrades done on specific routes, some of the classic routes have minimal drill bolts and only at difficult sections to not take the experience for the mountaineer

Authors’ response 3: Thanks for the constructive hint, unfortunately there was no information on that. Some alpinists try to climb something without artificial help whereas other use the drill bolts. Based on my experience, most alpinists use the drill bolts if available.

Reviewer 1 Comment 4 & 5: The anatomic location should be more specific. What do the authors mean by ankle vs ankle joint? Shank? Leg? Maybe something similar to below is more adequate? Similar for upper extremity and trunk. The anatomic location/injury needs to make sense! Arm?

Authors’ response 4 & 5: We appreciate your constructive critique and tried to implement it as good as possible. Some of the short reports written by members of the rescue organizations included only sparse information about the location of the injury not allowing further specification. We mentioned this in Table 2.

Reviewer 1 Comment 6 & 7: How many patients needed hospitalization? How many patients needed an operation?

Authors’ response 6 & 7: Indeed, these data would be valuable. Unfortunately, no information about hospitalization or surgery were available.

Reviewer 1 Comment 8: Where they any falls that could have been prevented? No equipment, wrong equipment etc.

Authors’ response 8: Yes, however this information was very fragmentary and was only available for a small number of case reports. In consequence, we allowed not picking up this point too much.

Reviewer 2

Reviewer 2 Comment 1: The article is a simple descriptive study which might allow useful formulation of public policy or approaches to rescue and therapy. It is worth publishing. Some of the translation into English should be improved for clarity.

Authors’ response 1: Thank you for this remark. We performed a thorough spelling/grammar check of the whole manuscript and believe that the readability has substantially improved.

Reviewer 2 Comment 2, 3 & 4: line 58 "absolved " would be better written as "successfully completed". line 60 "absolving " would be better written as "successfully completing" another option would be the climbing jargon " to summit". line 223 patients who "self confine" the best English equivalent would be patients who "self present" to general practitioners etc.

Authors’ response 2: We changed this accordingly.

Reviewer 2 Comment 5: In table 2 injuries to the trunk "rip" is not a common medical term. I don't know what that means. It may be a translation for laceration?

Authors’ response 3: Thank you for pointing this out. We were referring to an injury of the ribs/costae and corrected this mistake.

Reviewer 2 Comment 6: In the results section "risk aversion" is discussed. This is a psychological concept and may not explain the decrease in accident and injury rates. The change might be due to better training , better equipment or varying numbers of climbers going up mountains.

Authors’ response 4: Indeed, it is a psychological concept. We weakened the line of arguments (L150-152) and were reformulating the paragraph. As there might be – however strongly biased – a relationship between severity of an injury and risk aversion we did not delete the paragraph but made in the end another comment concerning a potential bias (L158-159).

---

## [Decision Letter · Decision Letter 1]

14 Mar 2022

4000ers in the Alps – so beautiful, so dangerous - an analysis of falls in the Swiss Alps between 2009-2020

PONE-D-21-24501R1

Dear Dr. Gasser,

We’re pleased to inform you that your manuscript has been judged scientifically suitable for publication and will be formally accepted for publication once it meets all outstanding technical requirements.

Kind regards,

Zsolt J. Balogh, MD, PhD, FRACS

Academic Editor

PLOS ONE

Additional Editor Comments (optional):

Thank you.

Reviewers' comments:

Reviewer's Responses to Questions

**Comments to the Author**

1. If the authors have adequately addressed your comments raised in a previous round of review and you feel that this manuscript is now acceptable for publication, you may indicate that here to bypass the “Comments to the Author” section, enter your conflict of interest statement in the “Confidential to Editor” section, and submit your "Accept" recommendation.

Reviewer #1: All comments have been addressed

Reviewer #2: All comments have been addressed

2. Is the manuscript technically sound, and do the data support the conclusions?

Reviewer #1: Yes

Reviewer #2: Yes

3. Has the statistical analysis been performed appropriately and rigorously? 

Reviewer #1: Yes

Reviewer #2: Yes

4. Have the authors made all data underlying the findings in their manuscript fully available?

Reviewer #1: Yes

Reviewer #2: (No Response)

5. Is the manuscript presented in an intelligible fashion and written in standard English?

Reviewer #1: Yes

Reviewer #2: (No Response)

6. Review Comments to the Author

Reviewer #1: All comments have been addressed sufficiently.............................

......................................................................................................................................................................................................................................................................................................................................................................................................................................................................................................................................................................................................................................................................................................................................................................................................................................................................................................................................................................................................................................................................................................................................................................................................................All comments have been addressed sufficiently.............................

......................................................................................................................................................................................................................................................................................................................................................................................................................................................................................................................................................................................................................................................................................................................................................................................................................................................................................................................................................................................................................................................................................................................................................................................................................

Reviewer #2: (No Response)

7. PLOS authors have the option to publish the peer review history of their article (what does this mean?). If published, this will include your full peer review and any attached files.

Reviewer #1: No

Reviewer #2: **Yes: **Conrad Loten

---

## [Editor Report · Acceptance letter]

28 Mar 2022

PONE-D-21-24501R1 

4000ers of the Alps – So beautiful, so dangerous: An analysis of falls in the Swiss Alps between 2009-2020 

Dear Dr. Gasser:

I'm pleased to inform you that your manuscript has been deemed suitable for publication in PLOS ONE. Congratulations! Your manuscript is now with our production department. 

Kind regards, 

on behalf of

Dr. Zsolt J. Balogh 

Academic Editor

PLOS ONE